# Microplastic-Contaminated Feed Interferes with Antioxidant Enzyme and Lysozyme Gene Expression of Pacific White Shrimp (*Litopenaeus vannamei*) Leading to Hepatopancreas Damage and Increased Mortality

**DOI:** 10.3390/ani12233308

**Published:** 2022-11-26

**Authors:** Songsak Niemcharoen, Thanida Haetrakul, Dušan Palić, Nantarika Chansue

**Affiliations:** 1Department of Veterinary Medicine, Faculty of Veterinary Science, Chulalongkorn University, Bangkok 10330, Thailand; 2Veterinary Medical Aquatic Animals Research Center of Excellence (VMARCE), Chulalongkorn University, Bangkok 10330, Thailand; 3Aquatic Resources Research Institute, Chulalongkorn University, Bangkok 10330, Thailand; 4Chair for Fish Diseases and Fisheries Biology, Faculty of Veterinary Medicine, Ludwig-Maximilians-University Munich, 80539 Munich, Germany

**Keywords:** high-density polyethylene, microplastics, nonspecific immune system, gene expression, histopathology, Pacific white shrimp

## Abstract

**Simple Summary:**

Plastic waste in the marine environment can be degraded into microplastic particles that are smaller than 5 mm in diameter. Such microplastic particles can interfere with the effectors of the nonspecific immune system in crustaceans, effectively debilitating a major defense mechanism against infectious diseases. Pacific white shrimp (*Litopenaeus vannamei*) is one of the most important marine crustacean species with global presence in the aquaculture industry that is at high risk of being contaminated with microplastics due to its feeding habits and benthic food sources. Such contamination may increase the risk of adverse effects on shrimp health. Thus, this study aimed to investigate the effects of oral intake of microplastic in food on the nonspecific immune system gene expression of Pacific white shrimp. We demonstrated immunosuppression at the gene expression level and histological damage of the hepatopancreas in shrimp fed with microplastic-contaminated food, leading to higher mortality. Information from *L. vannamei* can be used as a model system for other marine decapods. The increased presence of microplastic pollution in the environment and food has the potential to affect shrimp fisheries and aquaculture in the future, as well as further destabilize marine ecosystems already exposed to the combined anthropogenic and climate changes.

**Abstract:**

Microplastic pollution can interfere with aquatic animal health and nonspecific immunity, increasing the potential for pathogen infection in crustaceans. However, the long-term effects of microplastics on crustacean immunity are less understood, especially regarding their toxicity in Pacific white shrimp (*Litopenaeus vannamei*). Effects of high-density polyethylene microplastics (HDPE-MPs) in feed on the mortality rate, hepatopancreas, and nonspecific immune system gene expression of Pacific white shrimp are presented. The LC_50_ at day 28 of HDPE-MP exposure was determined as 3.074% HDPE-MP in feed. A significant upregulation of the superoxide dismutase (SOD) and glutathione peroxidase (GPx) genes was observed in shrimp that were fed with 0.1 and 0.5% of HDPE-MP; then, they were downregulated significantly, except for the SOD gene expression of shrimp fed with 0.1% of HDPE-MP. The lysozyme (LYZ) gene was upregulated significantly in shrimp that were fed with 0.5, 1, and 3% HDPE-MP for 7 days and downregulated significantly in HDPE-receiving groups for at least 14 days. Significant histopathological changes in the hepatopancreas were observed in the treatment groups. The histopathological score of each lesion was correlated with the increase in HDPE-MP concentration. This study shows that the ingestion of HDPE microplastics can alter the expression of nonspecific immune system genes and damage the hepatopancreas in Pacific white shrimp.

## 1. Introduction

The ubiquitous presence of plastic waste in marine environments is becoming a high-priority concern due to its persistence and adverse effects on marine and coastal species directly and indirectly [1]. Weathering effects on plastic debris at the interface of coastal and offshore marine environments produce particles between 1 µm to 5 mm in diameter, commonly referred to as “microplastic” [2]. Future projections for the accumulated mass of buoyant degraded microplastic materials from the ocean surface layer indicate that over one million tons of these particles will likely be polluting the oceans in the period from 2020 to 2030 [3]. This situation presents multiple new or undefined risks including the potential ingestion of microplastics by aquatic animals [4,5,6], possibly leading to microplastic particles contaminating the marine food chain, including seafood intended for human consumption [7].

One of the most abundant components of plastic waste, including degraded microplastics, is polyethylene (PE), which is a common plastic polymer used in various items such as plastic bags, plastic bottles, food packaging, clothing, ropes, nets, etc. PE plastic items are usually produced as single-use items, with a short useful life and quick transition to plastic waste, and are considered a significant source of microplastics in the environment. PE represents almost two-fifths of all plastic waste (38%) in the marine environment, out of which 21% is low-density polyethylene (LDPE) and 17% is high-density polyethylene (HDPE) [8]. PE microplastics can accumulate in the surface layers of sediments [9], increasing the likelihood of benthic species such as shrimps ingesting them accidentally due to their feeding behavior. PE plastic items are widely used in aquaculture production processes. Shrimp and other crustaceans are frequently raised in HDPE-lined ponds [10], and various plastic items such as anti-predator nets or mesh screens are made from polymers including PE and polypropylene (PP) and used to prevent undesirable organisms from entering the ponds [7]. Moreover, paddle-wheel aerators (usually made from HDPE) are intensively used in shrimp culture to increase dissolved oxygen in the ponds [11,12]. Therefore, plastic equipment routinely used in shrimp culture can act as a significant source of PE (including MP) in the aquaculture ponds, increasing the risk of PE contamination of seafood produced in such conditions.

Microplastics are increasingly being detected in the gastrointestinal tract and gills of various marine fauna, including crustaceans. A study on the langoustine (*Nephrops norvegicus*) found that microplastic fibers could accumulate in the stomach and result in false satiation and reduced feed consumption and body mass [13]. A study on shore crabs (*Carcinus maenas*) showed that polystyrene microspheres were retained in the foregut and on the external surface of the gills [14]. Moreover, the translocation of microplastics from the gastrointestinal tract to the internal organs was presented in decapods. Studies on two crab species, the fiddler crab (*Uca rapax*) and shore crab, demonstrated the translocation of microplastic to hemolymph and internal organs such as hepatopancreas, ovaries, and gills [15,16]. However, the effects of microplastics accumulated in the tissues were not mentioned in these studies. The study on Chinese mitten crabs (*Eriocheir sinensis*) by Yu et al. [17] demonstrated the accumulation of microplastics in hepatopancreas and noted changes in the enzyme activity and gene expression of superoxide dismutase (SOD), aspartate transaminase (GOT), glutathione (GSH), glutathione peroxidase (GPx), acetylcholinesterase, catalase (CAT), and alanine aminotransferase.

Pacific white shrimp (*Litopenaeus vannamei*) is one of the most important crustacean species in the global aquaculture industry and a significant source of seafood for people worldwide. In 2021, the world’s shrimp production is estimated to have reached over 4.5 million tons [18,19]. Considering that microplastic contamination in the fish meal showed 19% of PE microplastic [20] and that fish meal is an important ingredient of shrimp feed [21], it is likely that PE-contaminated shrimp feed may result in the ingestion and possibly bioaccumulation of microplastic particles and increased risk of adverse effects on the shrimp health in aquaculture. 

Since crustaceans do not have effectors of an acquired immune system similar to vertebrates, their defense mechanisms against pathogens rely almost exclusively on innate (nonspecific) immune responses such as anti-bacterial and antioxidant enzyme activity [22]. An anti-bacterial enzyme such as lysozyme (LYZ) responds against the infection of *Vibrio harveyi*, *V. campbellii*, *V. penaecida*, *V. parahaemolyticus*, and white spot syndrome virus (WSSV) [23,24,25,26]. Antioxidant enzymes such as superoxide dismutase (SOD) and glutathione peroxidase (GPx) act against reactive oxygen species (ROS) produced by *V. parahaemolyticus* and WSSV [23,27,28]. 

Microplastics can interfere with nonspecific immune function in crustaceans: studies on Chinese mitten crabs [17,29] and penaeid shrimp [30] found that microplastics alter the expression of enzymes and genes related to immune responses. Microplastics can also cause histopathological lesions of the hepatopancreas of Pacific white shrimp [31]. Therefore, *L. vannamei* can be used as a reliable model for ecotoxicological investigations in marine decapods. However, our understanding of the connection between immunosuppression and tissue damage in penaeid shrimp is not complete. The aim of the presented studies was to determine the effects of dietary HDPE microplastics (HDPE-MPs) on white-legged Pacific shrimp health, histopathology, antioxidant, and innate immune gene responses. 

## 2. Materials and Methods

### 2.1. Animal Use Protocol

Pacific white shrimp (*Litopenaeus vannamei*) were obtained from a shrimp farm in Phetchaburi province (Phetchaburi, Thailand). The average weight and total length of shrimp used in this study were approximately 4 g and 7 cm, respectively. The shrimp’s general health and presence of external parasites were checked, and the shrimp were quarantined in a 500 L tank with aerated seawater of 5 g L^−1^ salinity for 2 weeks. The stock density was set at 70 shrimp per m2 of water surface, 20% water change was performed twice a week, and the water quality parameters were monitored daily. During the experiment, ten shrimp were reared in a glass tank filled with 45 L of seawater at a salinity of 5 g L^−1^. Tanks were installed with an undergravel filtration system and aerated for 24 h. The water was changed by 20% twice a week, and water quality was monitored every day The shrimp were fed at the rate of 4% of their total body weight per day, divided into 3 feedings/day with commercial shrimp feed (Charoen Pokphand Foods, CP). The IACUC protocol number of this study is 2031077.

### 2.2. Phase 1: Toxicity Test

#### 2.2.1. Feed Preparation

Shrimp feed pellet was prepared from commercial shrimp feed powder (Charoen Pokphand Foods, CP; Appendix A) mixed with test concentrations of pure HDPE-MP with a diameter of 50 µm (Cospheric^TM^, Santa Barbara, CA, USA) and 3% wheat gluten, using the extruder machine with a 2 mm sieve. Pellets were dried in a hot air oven at 60 °C for 6 h. The feed pellets were divided into rations and stored at 4 °C until use. HDPE-MP was added as a percentage of feed (g kg^−1^) [32]. Five feeds were produced for the toxicity test: (A) control feed without HDPE-MP/0%; (B) feed with 0.5% HDPE-MP; (C) feed with 5% HDPE-MP; (D) feed with 10% HDPE-MP; (E) feed with 20% HDPE-MP. 

#### 2.2.2. HDPE-MP Exposure Scheme and Determination of LD_50_

Several factors can influence the toxicity of microplastic (shape, size, polymer type, presence of chemical additives, etc.) [33,34,35,36,37]. Therefore, a toxicity test was performed to determine the LD_50_ of HDPE-MP used in this study. Shrimp were divided into 5 groups of 10 shrimp each in triplicate tanks (30 shrimp/group). Each group was randomly assigned a different dose (0, 0.5, 5, 10, and 20%) of HDPE-MP and fed for 28 days. Shrimp were reared in a 60 L glass tank with a filtration system and continuous aeration [38]. Probit analysis of recorded mortality rates was used to calculate LD_50_ at day 28 after exposure with SPSS version 22 (IBM^®^). 

### 2.3. Phase 2: Determination of Gene Expression and Histopathology

#### 2.3.1. Feed Preparation

A shrimp feed pellet was prepared with the same method as in the toxicity test but with a different percentage of HDPE-MP. Five feeds were produced for determination of gene expression and histopathology: (A) control feed without HDPE-MP/0%; (B) feed with 0.1% HDPE-MP; (C) feed with 0.5% HDPE-MP; (D) feed with 1% HDPE-MP; (E) feed with 3% HDPE-MP. The highest concentration of HDPE-MP in feed (3%) was selected based on the calculated LD50 on the 28th day, and lower concentrations (0.1, 0.5, and 1% HDPE-MP) were designed based on the previous toxicity reports of microplastic in finfish diets [32,39,40]. Therefore, the highest concentration of HDPE-MP in this study is not representative of the actual level of microplastic that can be found in aquafeed.

#### 2.3.2. HDPE-MP Exposure Scheme

The LD_50_ of HDPE-MP was used to determine the sublethal doses fed to shrimp in this experiment. Shrimp were randomly selected from a 500 L stock tank and divided into 5 groups (negative control and 4 treatments) of 10 individuals in each of the three replicate aquaria (60 L, with husbandry conditions as described above). Different amount of HDPE-MP was added to feed for each group (0/control, 0.1, 0.5, 1, and 3%), and shrimp were fed for 28 days.

#### 2.3.3. Gene Expression Assay

The hepatopancreas was used to determine the gene expression in this study, since the microplastic was previously reported to interfere with various gene expression and enzyme activities of hepatopancreas [17,31,41]. Moreover, the hepatopancreas is sensitive to oxidative stress [42,43], one of the known toxic effects of microplastic [31,44,45,46]. Three shrimp from each group were randomly sacrificed for hepatopancreas collection on days 7, 14, 21, and 28. Hepatopancreas was preserved in QIAzol lysis reagent (Qiagen^®^, Düsseldorf, Germany) and stored at −80 °C, before homogenization with a vortex mixer and RNA extraction with RNeasy Mini kit (Qiagen^®^ Germany) and QIAzol lysis reagent. The reverse transcription was performed with QuantiNova reverse transcription kit (Qiagen^®^ Germany). The real-time polymerase chain reaction was performed with the QuantiNova SYBR Green PCR kit (Qiagen^®^ Germany). The expression of superoxide dismutase (SOD), glutathione peroxidase (GPx), and lysozyme (LYZ) genes were measured in this study. The glyceraldehyde 3-phosphate dehydrogenase (GAPDH) gene was used as a reference gene [47]. Specific primers were used for each gene (Table 1). All samples were analyzed in triplicate, using Rotor-Gene Q (Qiagen^®^) with the following thermocycling profile: 95 °C for 2 min of initial activation, followed by 45 cycles of denaturation at 95 °C for 10 s, annealing at 60 °C for 30 s and with extension at 72 °C for 30 s. Relative gene expression was calculated by using the 2^−ΔΔCt^ method [48].

#### 2.3.4. Histopathology

Two shrimp from each group were sacrificed on day 28 of microplastic exposure. The shrimp were individually preserved in Davidson’s solution before histology processing. The samples were prepared with a thickness of 5 µm and stained with H&E stain as in Gonçalves et al.’s [50] method. Histopathology lesions were observed through a light microscope (Olympus, CX21F21, 40×). Histopathological scoring was based on the amount of area that histopathological lesion occupied in the medium power field (40x) as in Littik’s [51] method. The scale indication of the criteria of the histopathological score is shown in Table 2.

#### 2.3.5. Statistical Analysis

Gene expression and mean histopathology score of each lesion were considered dependent values. Data between groups were evaluated by one-way analysis of variance (ANOVA) with Duncan’s MRT post hoc test. Results were considered significant if *p* < 0.05, and statistical analysis was performed using SPSS^®^ version 22 (IBM^®^). 

## 3. Results

### 3.1. Mortality Rate and HDPE-MP LD_50_

LD_50_ at day 28 of HDPE-MP ingestion was determined to be 3.074% HDPE-MP. Shrimp provided with 20% HDPE-MP started dying on day 3, with 100% mortality with 10 and 20% HDPE-MP on days 19 and 13, respectively (Figure 1A). The mortality rate was positively correlated with increases in HDPE microplastic concentration.

### 3.2. Nonspecific Immunity Gene Expression

Significant upregulation of the SOD gene was observed in shrimp provided with 0.1% HDPE MP for 14 days and shrimp provided with 0.5% HDPE-MP for 14 and 21 days (Figure 1B). SOD gene expression of 0.5, 1, and 3% HDPE-MP was downregulated significantly at day 28 (Figure 1B). GPx gene expression was upregulated significantly in shrimp provided with 0.1 and 0.5% HDPE-MP for 21 and 14 days, respectively (Figure 1C). GPx gene expression of shrimp provided with 0.1, 0.5, and 1% HDPE-MP for 28 days was downregulated significantly, while shrimp provided with 3% HDPE-MP were downregulated significantly at days 14, 21, and 28 (Figure 1C). LYZ gene was upregulated significantly in shrimp provided with 0.5, 1, and 3% HDPE-MP for 7 days and downregulated significantly in all groups provided with HDPE-MP for at least 14 days (Figure 1D).

### 3.3. Histopathology Lesions

Histopathological lesions were not observed in the hepatopancreas of the negative control group (Figure 2A). In contrast, the hepatopancreas of treatment groups presented with histopathological lesions, including interstitial hemocyte infiltration (Figure 2B), epithelium hyperplasia (Figure 2B,C), tubular deformity (Figure 2C,D), nodule formation (Figure 2E), and melanization (Figure 2E). Shrimp fed with 3% had a significantly higher histopathological score for interstitial hemocyte infiltration. Scores for shrimp fed with other concentrations of HDPE-MP were significantly different from the negative control group but not significantly different from each other (Figure 3). Shrimp provided with the lowest concentration of HDPE-MP (0.1% HDPE-MP) had the highest histopathological score of tubular hyperplasia. Shrimp provided with higher concentrations were not significantly different from the negative control group. There was no significant difference in the histopathological score of tubular deformity among the groups provided with HDPE-MP at different concentrations. Shrimp provided with at least 0.5% HDPE-MP had a significantly higher histopathological score of nodule formation than the negative control group. Only the shrimp provided with the highest HDPE-MP concentration (3% of HDPE-MP) had a significantly higher histopathological score of melanization than the negative control groups. All histopathological scores were increased corresponding to increased HDPE concentration, except tubular hyperplasia, which had a lesser histopathological score in shrimp provided with higher concentration (Figure 3).

### 3.4. Observation Finding Note

During the toxicity test, one of the deceased shrimp that were fed with 20% HDPE had intestine obstruction by a clump of HDPE-MP when observed under a light microscope (Figure 4A). The excreta of the shrimp were also collected and observed under a light microscope, and HDPE-MP was rediscovered in the excreta of shrimp that were fed with HDPE-MP (Figure 4B).

## 4. Discussion

A direct correlation between mortality rate and concentration of HDPE-MP was demonstrated. However, it is known that many factors can contribute to the toxicity of microplastic (including size, shape, polymerization, and chemical additives), which were previously described to affect ingestion, gut retention, and mortality rate [36,37,52,53,54]. For example, the impact of microplastic-contaminated water on the mortality rate and retention time in the gastrointestinal tract and gills of Daggerblade grass shrimp (*Palaemonetes pugio*) was influenced by microplastic particle shape and size [55]. Further, microplastics in the marine environment can also absorb toxic contaminants from their surroundings [56,57,58,59,60,61]. Ingestion of such microplastic fomites and their exposure to changes in the digestive tract environment (pH, enzymes) can cause the leaching of attached toxicants in the gut lumen and entering blood circulation [62,63,64,65,66], possibly exercising synergistic effects resulting in increased microplastic toxicity [32]. Thus, the LD_50_ of the microplastic particulate contaminants in the natural environment could be lower than determined here, indicating that a thorough toxicological risk assessment supported by relevant data is needed to estimate the impact of these novel contaminants.

Expression of the SOD and GPx genes at day 21 followed the gene expression pattern reported in mitten crab studies by Yu et al. [17]. Mitten crab exposure to polystyrene microplastic at low concentration (≤4 mg L^−1^) for 21 days upregulated SOD and GPx gene expression, while this effect was not noticed when a high dose of microplastic (40 mg L^−1^) was used. In our study, shrimp that ingested low concentrations of HDPE-MP (1 and 5% HDPE-MP) showed increased SOD and GPx gene expression compared to control, providing evidence of shrimp response to reactive oxygen species (ROS) or other forms of oxidative stress induced by HDPE-MP. This is in accordance with recent studies that demonstrated how the upregulation of antioxidant enzymes (SOD, GPx, catalase (CAT), and glutathione S-transferase (GST)) in animals exposed to microplastic was indicative of responding to the generation of ROS [17,31,67]. ROS increase is cytotoxic and can induce apoptosis in various organs [68], and the capacity of an organism for prevention of ROS-induced damage via antioxidant enzymes has limitations [69,70]. In our study, shrimp fed with a high concentration of HDPE-MP (1 and 3% HDPE-MP) for a longer time could not respond adequately to ROS, as evidenced by suppression of the expression of antioxidant enzymes. Such suppression resulted in excessive ROS damage to the hepatopancreas [71].

The alteration of antioxidant gene expression in this study could be also a consequence of stress response arising from ingestion of HDPE-MP. Stress factors such as abnormal temperature, pH changing, poor water quality, and starvation could induce oxidative stress and change antioxidant gene expression [43,72,73,74]. Dietary microplastic has been shown to induce starvation stress, as microplastic fibers disrupted feed ingestion and nutrient absorption, leading to starvation in the Norwegian lobster (*Nephrops norvegicus*) [13,75]. In the present study, during the toxicity test, obstructions of HDPE-MP in the intestine of one shrimp fed with 20% HDPE were observed under a light microscope (Figure 4A). This disruption of ingestion and nutrient absorption indicates that starvation stress might play a partial role in the alteration of antioxidant gene expression. It should be noted that in the present experiment, this was a single finding attributed to the nature of microplastic spheres, rarely found to accumulate in the gastrointestinal tracts of exposed animals [34,35,52]. The study by Lin et al. [74] indicated the downregulation of SOD gene expression to the lowest level within 5 days, in contrast with the upregulation (or absence of the downregulation) of the SOD gene expression within 7 days that is presented in this study. Thus, the result of gene expression in this study could be associated with the toxicity of microplastics instead of starvation, which implied the blockage of the intestine. Nevertheless, the HDPE-MP was routinely observed under a light microscope in the excreta of shrimp fed with HDPE-MP (Figure 4B). Fecal HDPE-MPs were not intact, and some were broken into pieces, strongly suggesting that the gastric mill (tooth-like structure in the anterior stomach of shrimp) could break down HDPE spheres into smaller microplastic or even grind them to nano-plastic sizes, increasing the risk of translocation to hepatopancreas. A similar correlation was observed in Antarctic krill (*Euphausia superba*) by Dawson et al. [76]. The possibility of translocation of microplastic/nano-plastic into hepatopancreas, together with starvation stress occurrence, suggested that alteration of SOD and GPx gene expression was related to HDPE-MP exposure.

The LYZ gene expression in this study was significantly upregulated after ingesting at least 0.5% HDPE-MP for 7 days. The previous studies showed that short-term microplastic exposure could upregulate LYZ gene expression and enzyme activity [29,30]. Interestingly, our study also showed that ingestion of at least 0.1% HDPE-MP for 14 days could significantly downregulate LYZ gene expression. Since lysozyme is the vital defense mechanism against harmful bacteria such as *Vibrio* spp. [26,77,78], downregulation of LYZ gene expression can increase shrimp vulnerability to this pathogen that causes early mortality syndrome [79]. Current literature presents evidence of both reduction and increase in Lysozyme activity and gene expression when an organism is exposed to plastic micro/nano-particles [30,80,81,82], suggesting that variability or contradiction in reported responses can be attributed to differences in study methods including animal species, type/size of plastic particles, and other confounding factors in non-standardized studies. Therefore, the exact mechanisms and causes of the upregulation or downregulation of LYZ gene expression in shrimp exposed to HDPE-MP currently remain unknown.

The study of the acute effect of microplastics on hepatopancreas histology by Wang et al. [30] showed that fluorescent red polyethylene microspheres (FRPE) could induce deformation in hepatopancreatic tubular. The present study focused on the chronic (>7 days) effects of microplastics in the hepatopancreas, and similar changes to tubular deformity and other histopathology lesions were observed with increased severity. The intensity of histopathology lesions observed in this study behaved in a dose-dependent manner: tubular deformity, interstitial hemocyte infiltration, nodule formation, and melanization increased at higher concentrations of ingestible HDPE-MP. Epithelial hyperplasia in shrimp that consumed a high dose of HDPE-MP advanced into later stages of this lesion (such as nodule formation and melanization), causing a relative reduction in early-stage hyperplastic changes. 

Increased interstitial hemocyte infiltration, nodule formation, and melanization that presented in the hepatopancreas of shrimp fed with HDPE-MP in this study might be evidence of chronic inflammation [83,84] and translocation of HDPE-MP from the gastrointestinal tract into hepatopancreas. Although we did not focus on the translocation and accumulation of ingested HDPE-MP, previous studies in fiddler crab (*Uca rapax*) by Brennecke et al. [15,30] demonstrated that microplastics with a diameter between 180 and 250 µm (larger than HDPE-MP Ø 50 µm used in our study) were observed in hepatopancreas. This information, along with evidence of breaking down HDPE-MP (Figure 4B), supports the possibility of translocation of microplastics into hepatopancreas that occurred in our study. Increased hepatopancreas nodule formation in shrimp exposed to HDPE-MP, together with other pathological responses indicative of chronic inflammation, suggests the possibility of reactive inflammation against HDPE-MP in hepatopancreas tissue, with subsequent severe tissue damage after prolonged exposure [22,85]. Such sustained damage leads to cellular dysfunction and can result in a decrease in nonspecific immunity gene expression observed in this and other studies [86]. Microplastic damage to the hepatopancreas could be the leading cause of mortality in this study, considering its essential role in digestion and metabolism regulation in crustaceans [87]. As mechanisms of microplastic recognition by the immune system in shrimp are not fully understood, further studies are necessary to support this hypothesis [88,89].

Microplastic effects on the antioxidant and lysozyme of *L. vannamei* with the different routes of administration and duration of exposure were reported. Wang et al. [30] immersed shrimps in seawater contaminated with polystyrene microspheres (Ø 5 µm) for 48 h and observed upregulation of lysozyme gene expression which correlated with the result of this study. Effects of polyethylene microplastics on oxidative stress were reported by Hsieh et al. [31] Shrimp were directly injected with fluorescent red polyethylene microspheres or FRPE (Ø 10–22 µm) intramuscularly. The results show significant upregulation of SOD and GPx gene expression at 7 days post initial injection, except in shrimp injected with the highest dosage of FRPE that showed significant downregulation of GPx gene expression. The results of SOD and GPx gene expression in our study are not significantly different from the negative control on day 7 of the experiment. However, the result of Hsieh et al. [31] indicated a pattern of SOD and GPx gene expression similar to this study, as well as the study in mitten crab by Yu et al. [17]. In all of the above reports, lower concentrations of microplastic were shown to upregulate the antioxidant genes, while the highest concentration of microplastic could downregulate the antioxidant genes, indicating a bi-phasic response possibly caused by depletion of the antioxidant capacity of animals exposed to a high concentration of microplastic.

Considering the MP toxicity presented in this study, estimates of average microplastic contamination in shrimp feed are 39 mg kg^−1^ of feed. Calculations were based on the estimated weight of microplastic particles of 0.02 mg and using 9% of fish meal [39,90] and the assumption that feed ingredients (fish meal and soy meal) were contaminated with microplastic in the worst-case scenario [91,92]. This estimate was lower than the concentration of HDPE-MP in this study; therefore, the result of this study was not representative of the actual level of microplastic that can be found in aquafeed today. However, as a significant increase in microplastic contamination in the environment is predicted [3], microplastic contamination level in the aquafeed will likely reflect this increase with a negative impact on Pacific white shrimp culture.

## 5. Conclusions

The presented results strongly suggest that ingestion of HDPE-MP with a diameter of 50 µm is related to significant mortality of Pacific white shrimp, with the determined LD_50_ at day 28 of 3.074% HDPI-MP. Shrimp receiving a sublethal dose of HDPE-MP for 28 days showed significantly altered expression of antioxidant enzyme genes and lysozyme, considered critical components of shrimp disease resistance and innate immunity. Pathology of hepatopancreas positively correlated with the dose and duration of HDPE-MP exposure. In conclusion, long-term ingestion of microplastics even in amounts below LD50 can cause increases in shrimp mortality. Dietary microplastics interfere with lysozyme gene expression and resilience to oxidative stress, contributing to increased pathogen susceptibility in exposed animals. The increased presence of microplastic pollution in the environment and food has the potential to affect marine shrimp, including increased risks of fishery and aquaculture production losses in the future.

## Figures and Tables

**Figure 1 animals-12-03308-f001:**
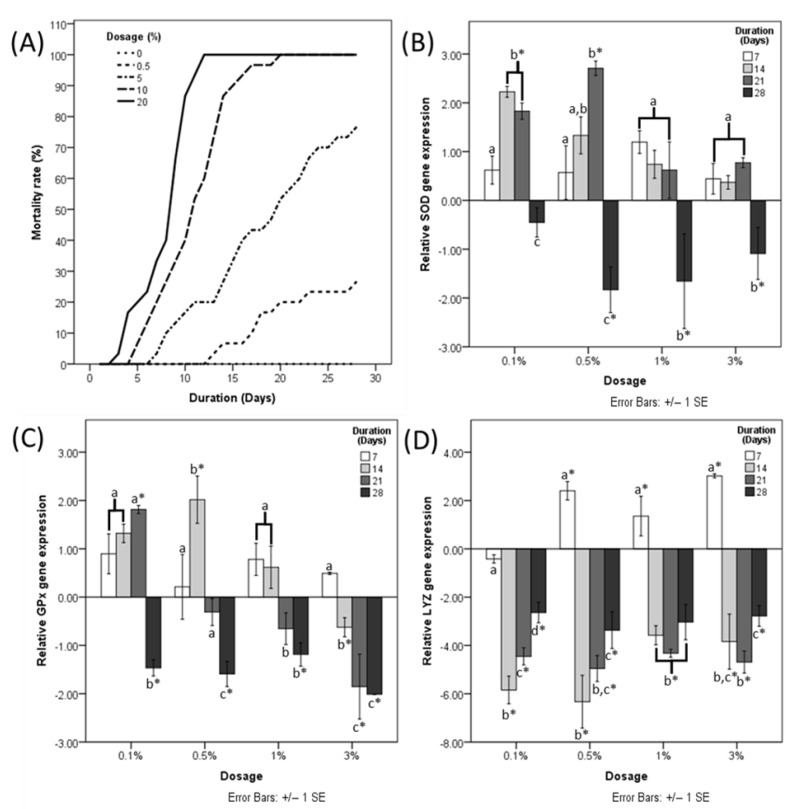
(**A**) Mortality rate of shrimp fed with different doses of HDPE microplastic. Fold change of (**B**) SOD, (**C**) GPx, and (**D**) LYZ gene expression normalized to a negative control at four different time points. Different letters (a, b, and c) above bars belonging to the same data series indicate significant differences between sampling times for each treatment. Asterisk indicates a significant difference from the negative control. *p* < 0.05.

**Figure 2 animals-12-03308-f002:**
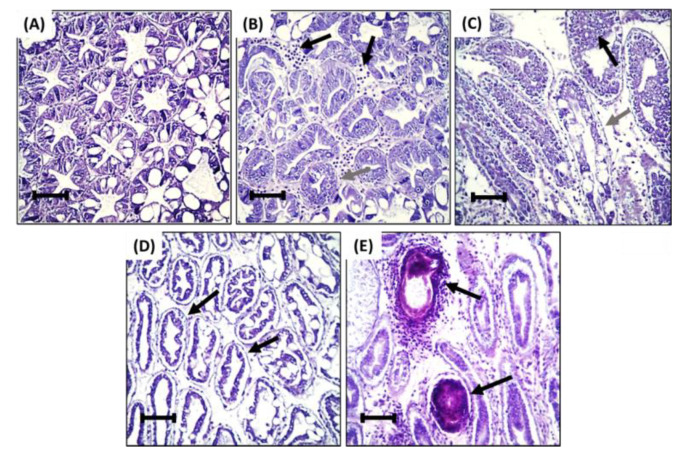
Representative histopathological findings from: (**A**) Normal hepatopancreatic tubule of negative control group. (**B**) Hepatopancreatic tubule of shrimp fed with 0.1% HDPE-MP with hemocyte between interstitial tissue of hepatopancreas (black arrow) and epithelium hyperplasia (gray arrow). (**C**) Hepatopancreatic tubule of shrimp fed with 0.5% HDPE-MP with epithelium hyperplasia (black arrow), epithelium detachment, and deformation of hepatopancreatic tubule (gray arrow). (**D**) Hepatopancreatic tubule of shrimp fed with 1% HDPE-MP with pronounced epithelium detachment and deformation of hepatopancreatic tubule (black arrow). (**E**) Hepatopancreatic tubule of shrimp fed with 3% HDPE-MP with nodule formation and melanization (black arrow). H&E staining, bar = 100 µm.

**Figure 3 animals-12-03308-f003:**
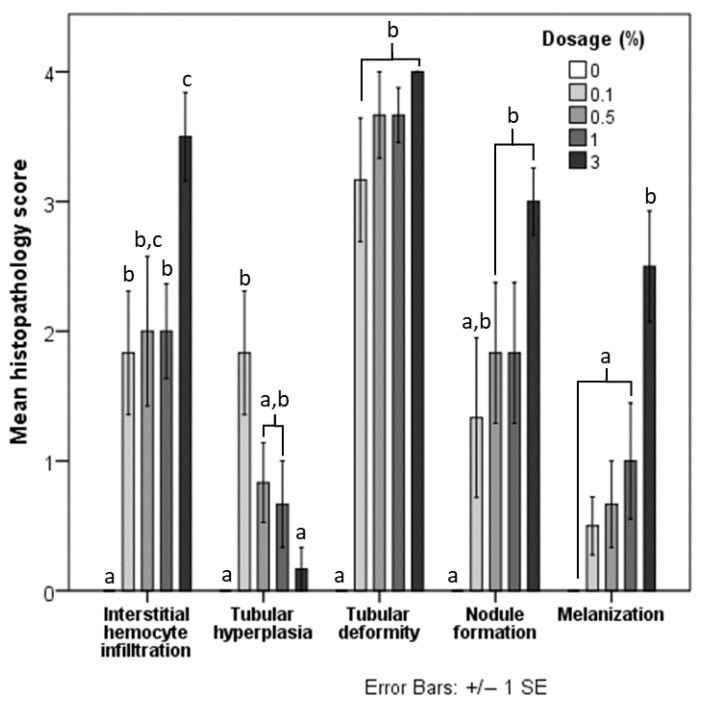
Mean histopathological score of each lesion in shrimp fed with five different doses of HDPE microplastics. Different letters (a, b, and c) above bars of the same series indicate significant differences in the mean histopathological score between the different doses (*p* < 0.05).

**Figure 4 animals-12-03308-f004:**
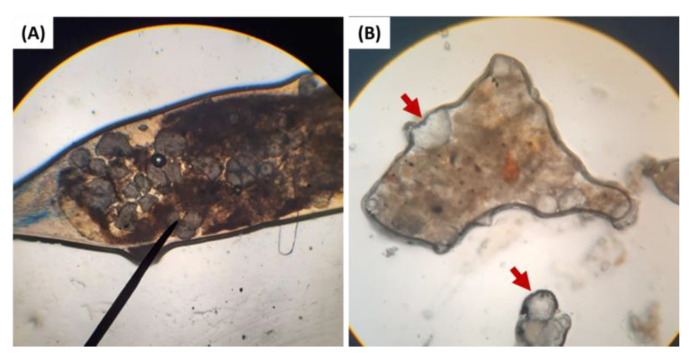
(**A**) The obstruction of HDPE-MP in the intestinal tract of shrimp that were fed with 20% HDPE. (**B**) HDPE-MP in excreta of HDPE-MP-fed shrimp. The red arrow indicates HDPE particles in excreta.

**Table 1 animals-12-03308-t001:** Genes and primers for each gene used in RT-qPCR.

Genes	Primer Sequence (5’-3’)	Tm (°C)	Product Size	NCBI Ref.
SOD	F-ATCCACCACACAAAGCATCA	55.9	229 BP	XM_027352840.1 [49]
	R-AGCTCTCGTCAATGGCTTGT	56.2		
GPx	F-TTTTTCCGTGCAAAAAGGAC	56.5	239 BP	XM_027376216.1 [49]
	R-TAATACGCGATGCCCCTAAC	56.4		
LYZ	F-GAAGCGACTACGGCAAGAAC	56.3	213 BP	XM_027372127.1 [49]
	R-AACCGTGAGACCAGCACTCT	56.0		
GAPDH	F-AAAGGTAGGAATTGCCCCCG	60.9	169 BP	XM_027372388.1
	R-GAAAGGGATGAGACTAGCACGAC	58		

Abbreviation: SOD = Superoxidase dismutase, GPx = Glutathione peroxidase, LYZ = Lysozyme, GAPDH = Glyceraldehyde 3-phosphate dehydrogenase.

**Table 2 animals-12-03308-t002:** Criteria of histopathological scoring.

Score	Indication
0	No histopathological lesion in any field on the slides
1	Histopathological lesion present in <25% of the fields on the slides
2	Histopathological lesion present in between 25% and 50% of the fields on the slides
3	Histopathological lesion present in between >50% and 70% of the fields on the slides
4	Histopathological lesion present in >75% of the fields on the slides

## Data Availability

Not applicable.

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
