# Peer review of "Microplastic-Contaminated Feed Interferes with Antioxidant Enzyme and Lysozyme Gene Expression of Pacific White Shrimp (Litopenaeus vannamei) Leading to Hepatopancreas Damage and Increased Mortality"

_animals, 2022, doi:10.3390/ani12233308_

Round 1

Reviewer 1 Report

The manuscript titled " Microplastics-contaminated Feed Interferes with Antioxidant  Enzyme and Immune Responses of Pacific white shrimp (Li- 3 topenaeus vannamei) leading to Hepatopancreas Damage and 4 Increased Mortality” investigated the effects of high-density polyeth-ylene microplastics (HDPE-MP) in feed on nonspecific immune system gene expression and histopathological changes in the hepatopancreas of Pacific white shrimp. It is interesting, but some issues should be addressed before acceptance. As a major revision, the manuscript needs to be reviewed by an English-language editor or proofreading service, the title and conclusion should reconsidered. Major points:

1. Line 35 “The LC50 at day 28 of HDPE-MP was 3.074% HDPE-MP”, what dose this sentence mean.

2. The author only examined the expression pattern of LYZ, however, this result is not enough to conclude that microplastics-contaminated feed interferes with immune responses of Pacific white shrimp.

3. The Latin of species should to be italicized.

4. Line 111, “in a 500 L tank with aerated seawater of 5 g L-1 salinity for 2 weeks”, “5 g L-1 salinity” it's not the normal seawater concentration, why?

5. Table 3, all primers stated here were form reference 37?, If yes, it’s not necessary to show the primer sequences here.

6. Figure 1 and 3, the significant differences symbol annotation here were incorrect.

7. We cannot find Figure 4

Author Response

Dear Reviewer

Thank you for your comments on this manuscript. We provide the answers to your questions and comments, Please see the attachment. 

Reviewer 2 Report

The manuscript examine the effect of microplastics-contaminated feed on juvenile Pacific white shrimp. The topic is highly relevant to the global aquaculture industry and the effect of microplastic pollution in the environment on aquatic animals. Overall, the manuscript was quite good. There is some minor issue with the writing (such as word choice) and crucial explanation in certain area is needed. 

(1) Title: the title of the manuscript may need to be adjusted to better match the content. The manuscript examined the gene expressions of the antioxidant enzyme and immune responses, but not measuring the actual enzyme nor immune activity. Therefore, the current title may mislead reader to think that actual activities were measured. Please consider adjusting the title and add "gene expression".  

(2) Title: While the increased in the concentration of microplastics resulted in higher mortality in the toxicity test, claiming "increased mortality" seemed overreaching to me. The concentration for toxicity test is chosen with the intention to cause mortality, and higher concentration will result in higher mortality.  If there is additional survival rate results of Phase 2 to support the increased mortality claim, then that would be more appropriate. Otherwise, it is probably overreaching to claim "increased mortality".

(3) Additional details on the existing reports on microplastic in shrimp could be added. Particularly, those that highlight the difference of the current experiment compared to others such as the size of the shrimp used, the method (ingestion with dietary MP vs. exposure of MP in water), other toxicity studies. 

(4) The choice of MP concentration in the second experiment. How are these concentration chosen? Also, what is the actual level of MP that can be found in aquafeed? How realistic is the concentration in this experiment? Please explain and provide the justification in the methods as well as the discussion on the practical aspect of this findings. You may provide an estimated value if there's no information, but a discussion on the MP concentration have be added. (Please see: Lenz, R., Enders, K., & Nielsen, T. G. (2016). Microplastic exposure studies should be environmentally realistic. Proceedings of the National Academy of Sciences, 113(29), E4121-E4122.)

Author Response

(The authors gave the same response as above.)

Reviewer 3 Report

The paper entitled Microplastics-contaminated Feed Interferes with Antioxidant 2 Enzyme and Immune Responses of Pacific white shrimp (Li-3 topenaeus vannamei) leading to Hepatopancreas Damage and 4 Increased Mortality Niemcharoen, S et al., is very interesting. This study is very important at present, as there is much concern regarding microplastic accumulation in marine animals.  

Introduction

Make a note of any previous instances of detecting microplastics in decapod crustaceans if any.  If so, what are the possible organs of accumulation and effects? 

Materials and methods 

Please provide hydrological conditions at which shrimps are maintained 

Line 107: Species must be in name italics 

Line 112: m2 correct as m2

Line 136-137: Mention the software used for Probit analysis

Gene expression assay: Explain the rationale behind the selection of hepatopancreas as the target tissue for gene expression

Discussion:

The results suggest that feed digestion in shrimp's intestines is impaired.  Thus leading to starvation effects. Under such conditions how gene expression patterns can be differentiated between toxicity and starvation effects.  Discuss this possibility 

Suggestions:  It is interesting to study the amount of microplastic accumulated in the shrimp organs and correlate with the gene expression and histology studies 

Also, it is interesting to do the histology and gene expression of shrimp intestines concerning microplastic accumulation  

Author Response

(The authors gave the same response as above.)

Round 2

Reviewer 1 Report

No more comments.